# Effects of Starter Diet Energy Concentration on Nutrient Digestibility and Subsequent Growth Performance and Meat Yields of Broilers under Two Coccidiosis Control Programs

**DOI:** 10.3390/ani14111524

**Published:** 2024-05-22

**Authors:** Alyson G. Myers, Samuel J. Rochell

**Affiliations:** 1Department of Poultry Science, University of Arkansas, Fayetteville, AR 72701, USA; 2Department of Poultry Science, Auburn University, Auburn, AL 36849, USA

**Keywords:** coccidiosis, broiler, nutrition, energy, lipid, digestibility

## Abstract

**Simple Summary:**

Simple Summary: Coccidiosis vaccination is a widely used and effective tool for protecting poultry against clinical coccidiosis outbreaks. However, live *Eimeria* vaccines induce mild, transient epithelial damage and inflammation as immunity develops during oocyst cycling. This process can impair nutrient utilization, particularly for dietary lipids. Therefore, this experiment studied the influence of dietary energy level (standard, moderate, or high), adjusted by varying soybean oil inclusion, during the starter phase (0 to 18 d post-hatch) on nutrient and energy digestibility and subsequent performance and carcass characteristics of broilers either vaccinated against coccidiosis on day of hatch or not vaccinated and provided an in-feed anticoccidial drug. Vaccination impaired nutrient and energy digestibility, weight gain up to 31 d, feed conversion, and carcass yield of broilers compared with those fed an in-feed anticoccidial. However, the impacts of vaccination on feed conversion ratio were dependent upon starter diet energy/lipid concentration, even during feeding phases after the starter period, with greater impacts observed at higher energy/lipid levels. Thus, compensating for impaired lipid and energy utilization by coccidiosis-vaccinated broilers through increasing dietary lipid content is an ineffective, and perhaps detrimental, strategy for vaccinated broilers.

**Abstract:**

An experiment was conducted to evaluate the live performance, processing characteristics, and apparent ileal digestibility (AID) of nutrients and energy (IDE) in broilers under two coccidiosis control programs (CCP) and fed three starter diet energy levels. Treatments were a factorial arrangement of CCP [in-feed diclazuril (ACD) or vaccinated after hatch (VAC)] and three starter diet energy levels [3008 (standard), 3058 (moderate), and 3108 (high) kcal/kg apparent ME_n_] achieved with different soybean oil concentrations. Birds were reared in floor pens (12 per pen) and received experimental starter diets from 0 to 18 d and common grower and finisher diets to 43 d. At d 11, VAC birds had higher (*p* < 0.05) excreta oocyst counts and lower (*p* < 0.05) plasma carotenoids, nutrient AID, and IDE than ACD birds. From 0 to 18 and 0 to 31 d, VAC decreased (*p* < 0.05) body weight gain and increased (energy × CCP, *p* < 0.05) feed conversion ratio of birds fed the moderate and high-energy diets but not for those fed the standard energy diet. From 0 to 43 d, VAC only increased the feed conversion ratio of birds fed the moderate-energy starter diet (energy × CCP, *p* < 0.05). Carcass yields were lower (*p* < 0.05) for VAC birds than for ACD birds, and interactive effects (*p* < 0.05) were observed for wing yield. In summary, increasing dietary lipid concentration to account for *Eimeria*-induced reductions in lipid digestibility during the starter period of coccidiosis-vaccinated broilers may exacerbate, rather than ameliorate, these impacts on bird performance.

## 1. Introduction

Consumer demand for poultry products originating from birds reared without antibiotics has increased the use of vaccines to replace ionophore anticoccidial drugs for the control of coccidiosis. As such, vaccines are increasingly used in rotational or “bioshuttle” programs with chemical anticoccidial drugs, which are often permitted in these systems but are susceptible to resistance. Although live oocyst vaccination is effective in promoting immunity and protection against clinical coccidiosis, it can induce a mild transient form of coccidiosis, referred to as “coccidiasis”, that usually occurs between 14 and 28 d post-hatch impairs broiler performance [1]. Impaired feed efficiency during vaccine cycling is presumably due in part to nutrient malabsorption associated with intestinal damage and inflammation attributed to the sub-clinical, vaccine-induced infection [1,2,3].

Previous work conducted in our laboratory longitudinally characterized nutrient digestibility in floor-reared broilers given a coccidiosis vaccine on the day of hatch and showed that nutrient digestibility was impaired the most at 12 d post-hatch, with the greatest duration and magnitude of this impact occurring for lipid digestibility [4]. Impaired lipid digestibility also reduces the absorption of fat-soluble nutrients, including vitamins and carotenoids, which may lead to indirect consequences on bone health and pigmentation [5]. Intestinal damage associated with an *Eimeria* infection also increases epithelial cell turnover and induces an inflammatory immune response, both of which are energetically costly and occur at the expense of broiler performance [6,7]. Indeed, the energetic costs of impaired digestibility and metabolic changes during a coccidial infection are consequential [8].

Providing an increased concentration or a more available source of nutrients for which digestibility is impaired is a potential strategy to support the performance of broilers during coccidial vaccine cycling. For example, previous research reported that increased dietary digestible amino acid content improved the performance of broilers challenged with a coccidiosis vaccine [3,9,10]. Korver et al. [11] reported that increasing dietary metabolizable energy from 2714 to 3303 kcal/kg by increasing corn oil supplementation improved the performance of lipopolysaccharide-challenged broilers, whereas increased tallow supplementation did not, indicating that both dietary lipid type and level may influence inflammatory responses. However, to our knowledge, responses of coccidiosis-vaccinated broilers to increased dietary energy resulting from higher lipid inclusion have not been investigated. Therefore, the objective of this experiment was to test the hypothesis that live performance and processing characteristics (i.e., deboned parts yield) of coccidiosis-vaccinated broilers can be improved by increasing dietary soybean oil inclusion during the starter phase (0 to 18 d) to account for the expected vaccine-induced reductions in energy digestibility during this period.

## 2. Materials and Methods

### 2.1. General Bird Husbandry and Dietary Treatments

A total of 1368 male broiler by-product chicks from a Cobb 500 female line were obtained from a commercial hatchery on the day of hatch. Upon arrival, one-half (684) of the chicks was orally gavaged with the manufacturer’s recommended dose of a live oocyst vaccine (Coccivac^®^-B52; Merck Animal Health, Intervet Inc., Millsboro, DE, USA) (VAC), whereas the other half was not vaccinated and received in-feed diclazuril (Clinacox, Huvepharma, Peachtree City, GA, USA), a chemical anticoccidial drug (ACD), throughout the entire experiment. The vaccine was delivered by oral gavage (0.25 mL/bird) using a stainless-steel gavage needle to provide uniform administration. All chicks were group-weighed and distributed to 114 floor pens on unused pine shavings. Throughout the trial, litter was sprayed daily with water to increase litter moisture and promote oocyst sporulation. Each floor pen (0.91 × 1.22 m) was equipped with a hanging feeder and a nipple drinker line and contained 12 birds (0.09 m^2^ per bird). Birds were provided access to feed and water ad libitum throughout the 43 d experiment, and the lighting schedule and temperature targets were adjusted according to published management guidelines [12].

From 0 to 18 d of age, ACD and VAC birds were provided one of three isonitrogenous experimental starter diets that contained 2.10, 2.67, or 3.24% soybean oil (ME value of soybean oil used in dietary formulation was 8800 kcal/kg) included at the expense of cellulose, with calculated AME_n_ contents of 3008 (standard), 3058, (moderate) and 3108 (high) kcal/kg, respectively (Table 1). Throughout the remainder of the trial, ACD and VAC birds were fed common grower (18 to 31 d) and finisher (31 to 43 d) diets. Corn, soybean meal, and distiller’s dried grains with solubles were used as the primary ingredients, and all diets in Table 1 were formulated to meet or exceed published nutrient recommendations [13].

### 2.2. Determination of Growth Performance, Plasma Carotenoids, and Excreta Oocyst Shedding

Birds and feeders were weighed at 0, 11, 18, 32, and 43 d post-hatch for calculation of body weight gain (BWG), feed intake (FI), and feed conversion ratio (FCR) to assess growth performance. All dead and culled birds were weighed individually, and FCR calculations were adjusted to include the weight gain of dead birds. From 0 to 11 d post-hatch, performance parameters were represented by 19 replicate pens for each treatment.

At 11 d post-hatch, after all pens had been weighed, all birds in 7 replicate pens were euthanized for the collection of blood and ileal digesta, whereas the remaining 12 replicate pens were used for performance measurements throughout the remainder of the trial. All birds from each treatment’s seven replicate sampling pens were euthanized by CO_2_ inhalation. Blood was collected from two randomly selected birds per pen via cardiac puncture and placed into tubes containing EDTA. After collection, tubes were placed on ice and subsequently centrifuged for 15 min at 1300× *g* and 4 °C to separate plasma. Plasma from birds within a pen were pooled, aliquoted, and stored at −80 °C until further analysis. All blood processing and carotenoid analysis procedures were conducted under yellow light and were determined by spectrophotometry as previously described by Allen [14]. Duplicate aliquots of 0.1 mL of plasma from each bird were diluted in 0.9 mL of denatured ethanol, centrifuged at 3000 rpm for 10 min to sediment protein, and 200 µL of supernatant from each sample was transferred into duplicate wells of a microtiter plate. Absorbance at 530 nm was subtracted from absorbance at 474 nm. The difference in these values was multiplied by 81 as the extinction coefficient of lutein to calculate lutein equivalents.

Oocyst shedding, as indicated by the number of oocysts per gram of pooled excreta samples collected from each pen, was determined before bird placement and at 11 d post-vaccination to confirm vaccine cycling. Samples of unused litter shavings were taken before placement, whereas fresh excreta samples at 11 d were collected from each pen using wax paper placed on the litter 12 h prior to collection. Each sample consisted of multiple droppings pooled within pen. All samples were placed in airtight conical tubes and kept refrigerated until processing. Samples were soaked in water overnight and homogenized by vigorous stirring. Following homogenization, 1 mL of sample was further diluted with 9 mL of saturated salt solution and pipetted into the chamber of a McMaster counting slide. Duplicate counts were made for each sample using the following equation:Oocysts per gram of excreta = (Oocyst count × dilution × volume)/(volume of counting chamber × weight of sample)
where the dilution was 10, and the volume of the counting chamber was 0.15 mL.

### 2.3. Determination of Apparent Ileal Digestibility and Ileal Digestible Energy

Digesta contents of the distal half of the ileum from all birds in the 7 replicate pens used for sampling were collected by gently flushing with deionized water. Digesta samples within each pen were pooled and frozen (−20 °C) until analysis. Frozen digesta samples were lyophilized and finely ground using an electric coffee grinder. Diet and digesta samples were analyzed for dry matter, gross energy, nitrogen, and ether extract content. Gross energy was determined with a bomb calorimeter (Parr 6200 bomb calorimeter, Parr Instruments Co., Moline, IL, USA). Nitrogen was determined using the combustion method (Fisions NA-2000, CE Elantech, Lakewood, NJ, USA) standardized with EDTA (method 990.03, AOAC International [15]) and ether extract was determined according to AOAC [15] method 920.39. Titanium dioxide was included in the feed at 0.5% as an indigestible marker, and diet and digesta TiO_2_ concentrations were determined in duplicate following the procedures of Short et al. [16]. Apparent ileal digestibility (AID) of dry matter, gross energy, ether extract, and nitrogen were calculated using the following equation:AID, % = {[(X/TiO_2_)_diet_ − (X/TiO_2_)_digesta_]/(X/TiO_2_)_diet_} × 100,
where (X/TiO_2_) = ratio of nutrient concentration (%) to TiO_2_ (%) in the diet or ileal digesta. Energy digestibility (%) values obtained from the above equation were multiplied by the gross energy content of the feed to calculate apparent ileal digestible energy (IDE) in units of kcal/kg.

### 2.4. Processing Characteristics

At 43 d post-hatch, all birds from eight replicate pens were selected for processing on d 44 after 12 h of feed withdrawal. Birds were transported in coops to the University of Arkansas Pilot Processing Plant. Individual live bird weights were recorded immediately before live-hanging. Birds were humanely euthanized, processed, and eviscerated. Hot carcass and abdominal fat pad weights were collected, and carcasses were chilled for at least 2 h before deboning. All carcasses from each pen were deboned for collection of parts weights and yields, with yield calculated as a percentage of pre-slaughter live body weight. Processing outcomes included weights and yields of the following: hot and chilled carcass, hot abdominal fat pad, pectoralis major (breast), pectoralis minor (tenders), leg quarters (thigh and drum), and wings.

### 2.5. Statistical Analysis

Treatments were comprised of a factorial arrangement of coccidiosis control program (CCP; ACD or VAC) × 3 starter dietary energy levels in a completely randomized block design. The pen was considered the experiment unit for all measurements, and pen location was a random blocking factor. Oocyst count data were arcsin square root transformed to achieve normality prior to statistical analysis. Data within each time point were subjected to a two-way ANOVA using the MIXED procedure of SAS 9.4 to assess the main effects of dietary energy level, vaccination status, and their interaction. Statistically different main effect means were separated using a Tukey’s multiple comparison test. When interactions were present, single degree of freedom orthogonal contrasts were used to compare ACD and VAC birds within a dietary energy level, with differences based on this comparison denoted with an asterisk. Statistical significance was considered at *p* < 0.05 in all cases.

## 3. Results

### 3.1. Oocyst Shedding and Plasma Carotenoids

No oocysts were detected in the shavings before bird placement. At 11 d post-hatch, VAC birds had higher (*p* < 0.05) excreta oocyst output and lower (*p* < 0.05) plasma carotenoid concentrations than ACD birds, with no main or interactive effects (*p* > 0.05) of energy level (Table 2).

### 3.2. Growth Performance and Apparent Ileal Digestibility of Nutrients and Ileal Digestible Energy at 11 d Post-Hatch

Growth performance from 0 to 11 d post-hatch and nutrient and energy digestibility data are presented in Table 3. At 11 d, VAC birds had lower (*p* < 0.05) BWG and higher (*p* < 0.05) FCR than ACD birds, with no differences (*p* > 0.05) in FI. Main effects of CCP (*p* < 0.05) and energy level (*p* < 0.05) were observed for AID of nutrients and IDE at 11 d post-hatch. Vaccinated birds had a lower (*p* < 0.05) AID of dry matter, nitrogen, ether extract, and energy and IDE compared with ACD birds. The AID of nitrogen was influenced by energy level (*p* < 0.05) and was highest for the high-energy diet (81%), intermediate for the moderate-energy diet (80%), and lowest for the standard energy diet (78%). Energy level also influenced the AID of ether extract (*p* < 0.05) and was highest for the standard energy diet (91%), intermediate for the moderate-energy diet (88%), and lowest for the high-energy diet (86%). Furthermore, energy level also influenced IDE (*p* < 0.05) and was highest for the high-energy diet (3426 kcal/kg), intermediate for the moderate-energy diet (3384 kcal/kg), and lowest for the standard energy diet (3279 kcal/kg). Dietary energy level did not influence (*p* > 0.05) AID of dry matter or energy.

### 3.3. Growth Performance to 43 d Post-Hatch and Processing Characteristics

At 18 d post-hatch, VAC birds had lower (*p* < 0.05) BWG and FI than ACD birds, with no main or interactive effects of energy level (Table 4). An energy level by CCP interaction was observed for FCR, where FCR was increased (*p* < 0.05) by VAC in birds fed the moderate and high-energy diets but was not influenced by VAC (*p* > 0.05) in birds fed the standard energy diet. After 18 d post-hatch, birds were fed a common grower diet until 31 d post-hatch and a common finisher diet was fed from 31 to 43 d post-hatch. At 31 d post-hatch, BWG (*p* < 0.05) was lower for VAC birds than for ACD birds, with no dietary effects. There were no treatment effects on FI (*p* > 0.05), but similar to the starter period, an energy level by CCP interaction was observed for FCR (*p* < 0.05) whereby VAC increased FCR in birds fed the moderate and high-energy diets, but not those (*p* > 0.05) fed the standard energy diet. At 43 d post-hatch, neither CCP nor energy level impacted broiler BWG or FI (*p* > 0.05). However, an energy level by CCP interaction for FCR persisted whereby FCR was increased (*p* < 0.05) by VAC in birds fed the moderate-energy diet but was not influenced by VAC (*p* > 0.05) in birds fed the standard or high-energy diets.

At 44 d post-hatch, hot and chilled carcass and wing yields were lower (*p* < 0.05) for VAC birds than for ACD birds, with no effects of dietary energy (Table 5). A diet by CCP interaction for wing weight was observed, whereby VAC reduced (*p* < 0.05) the wing weight in birds fed the moderate and high-energy diets but not those (*p* > 0.05) fed the standard energy diet. No other processing measurement was influenced by energy level or CCP (*p* > 0.05).

## 4. Discussion

Coccidiosis vaccination with live *Eimeria* induces immunity in chickens after several rounds of oocyst cycling [17]. In the current experiment, increased excreta oocyst output confirmed vaccinal oocyst cycling and indicated a successful model for comparing the responses of birds to dietary energy under two coccidiosis control programs. Plasma carotenoids, a sensitive indicator of coccidial-induced intestinal damage in birds [18], reflected malabsorption in VAC birds compared with those given an ACD. Coccidiosis vaccination on the day of hatch also reduced BWG and impaired FCR of broilers at 11 d post-hatch, with no effects on FI. This impaired FCR was likely due in part to nutrient malabsorption, as vaccination decreased IDE by 303 kcal/kg and AID of dry matter, nitrogen, and ether extract by 5.7, 4.5, and 6.2 percentage units, respectively, when compared with ACD birds at 11 d post-hatch. This vaccine-induced decrease in nutrient digestibility, particularly for lipids (ether extract), was expected based on previous reports [19].

The magnitude of reductions in IDE associated with coccidiosis vaccination in previous experiments in our lab [4,20] informed the dietary energy concentrations evaluated in the current study. As the energy density of starter diets increased with soybean oil supplementation, the AID of nitrogen and IDE also increased. Likewise, previous research has shown that supplemental lipids can reduce the digesta passage rate through the intestinal tract, allowing more time for the digestion and absorption of all nutrients present in the diet, including protein and starch [21,22]. The reduction in AID of ether extract that occurred as soybean oil inclusion increased, regardless of vaccination status, may reflect a limited ability of broilers at this age to efficiently digest or absorb lipids beyond a certain threshold [23]. Furthermore, despite being fat-soluble components, plasma carotenoids were not influenced by soybean oil content, indicating that the amount of added lipid in the standard energy diet was likely not limiting carotenoid absorption for birds in either coccidiosis control program.

The reduction in BWG for VAC birds observed at 18 and 31 d diminished by 43 d post-hatch. Previous literature has also reported broilers administered an *Eimeria* vaccine at day of hatch had a 3 to 4% reduction in BWG at 17 and 21 d post-hatch, with no differences in BWG observed at 28 d post-hatch when compared with non-vaccinated broilers [4,24,25]. Therefore, as intended, coccidiosis vaccination provided early *Eimeria* exposure, which allowed birds to compensate for the minor reduction in weight before the end of the grow-out period. Similar to the final live weights, vaccination did not impact hot or chilled carcass weights or yields at 44 d post-hatch in the current experiment. The reduction in wing weight for VAC birds fed moderate and high-energy diets may be due to differences in the protein and amino acid composition of the wing compared with other processing parts, which were not impacted by energy level or vaccination status [9]. Starter diet energy levels did not influence any processing characteristics. Similarly, Birk et al. [26] reported no differences in processing yields at 50 d of age when broilers were fed pre-starter diets that contained increased energy densities achieved by increased lipid inclusions at 1.3, 6, or 8% of the diet.

Despite the lack of difference in final BW and processing yields, increasing the dietary energy density by soybean oil supplementation in the starter period negatively impacted the FCR of VAC birds, not only during the starter period but throughout the grower and finisher periods, which was contrary to our hypothesis. Vaccinated birds fed the moderate and high-energy diets during the starter period had impaired FCR at 18 and 31 d post-hatch, and while no differences in FCR were observed between ACD and VAC birds fed the high-energy diet at 43 d post-hatch, the FCR of VAC birds fed the moderate-energy diet remained impaired. A lack of interaction between starter energy level and CCP on 11 d nutrient digestibility indicates that causes other than altered digestibility were likely responsible for the persistent energy level by vaccine interaction for FCR.

The unexpected, relatively greater impact of coccidiosis vaccination on FCR of birds fed the moderate and high-energy levels than for those fed the standard energy diet may have resulted from a proportionately greater need by vaccinated birds for glucose than for lipids as a fuel source. Benson et al. [27] reported that BWG and FCR of lipopolysaccharide-injected birds responded positively when dietary energy was increased from 2800 to 3200 kcal ME/kg using cornstarch, whereas the same increase in energy achieved with corn oil exacerbated the growth depression caused by lipopolysaccharide injection. Other experiments using lipopolysaccharide as an inflammatory agent have demonstrated increased demand for glucose by peripheral tissues and enhanced rates of gluconeogenesis from various precursors [28].

Enteric pathogenic infection likely also increases the demand for glucose per se as a fuel source for enterocytes. Indeed, the demand for glutamine, a glucogenic amino acid, increases during coccidiosis vaccination, potentially for its role as an energy source for immune cells and enterocytes [29]. Additionally, serum lipid levels have been reported to increase following infection or inflammatory response, perhaps due to increased lipolysis and decreased lipoprotein lipase activity, which alters the uptake and utilization of fatty acids during an inflammatory response [30,31,32]. In humans, the consumption of high-fat diets has been shown to increase intestinal permeability via a reduction in mucin production and downregulation of tight junction proteins [33,34]. Therefore, the intestinal damage and inflammation associated with the sub-clinical, vaccine-induced infection in the current experiment may have been exacerbated when birds were fed increased amounts of soybean oil. As such, an increase in energy density by carbohydrate or amino acid supplementation, and not lipid, may be a more effective strategy for supporting the performance of coccidiosis-vaccinated broilers.

## 5. Conclusions

In conclusion, coccidiosis vaccination did not compromise final BWG, FI, or most processed parts weights of 43 d old broilers, and a lack of response in these measurements to dietary energy during the starter period (0 to 18 d) indicates that this was not limiting dietary component for these outcomes. However, contrary to our original hypothesis, feeding increased energy density diets through greater soybean oil inclusion during the starter period caused detrimental effects on the feed efficiency of vaccinated broilers that persisted beyond the period of vaccine cycling. As such, further research is warranted to understand the mechanisms of *Eimeria*-induced lipid malabsorption and its impacts on broiler gastrointestinal health beyond reduced energy availability and lipid-soluble vitamin absorption.

## Figures and Tables

**Table 1 animals-14-01524-t001:** Composition of diets fed to broilers from 0 to 43 d post-hatch ^1^.

Ingredient, % as-Fed	Experimental (0–18 d)			Common Grower (18–31 d)	Common Finisher (31–43 d)
	Standard	Moderate	High		
Corn	53.94	53.94	53.94	58.46	60.45
Soybean meal (45%)	34.81	34.81	34.81	30.12	27.84
DDGS	4.00	4.00	4.00	6.00	6.00
Soybean oil	2.10	2.67	3.24	2.29	2.97
Limestone	1.09	1.09	1.09	1.09	1.03
Dicalcium phosphate	1.03	1.03	1.03	0.83	0.63
Salt	0.34	0.34	0.34	0.33	0.31
DL-methionine	0.31	0.31	0.31	0.25	0.21
L-lysine HCl	0.23	0.23	0.23	0.18	0.10
L-threonine	0.09	0.09	0.09	0.06	0.08
Trace mineral premix ^2^	0.10	0.10	0.10	0.10	0.10
Vitamin premix ^3^	0.10	0.10	0.10	0.10	0.10
Se premix ^4^ (0.06%)	0.02	0.02	0.02	0.02	0.02
Choline chloride (60%)	0.10	0.10	0.10	0.05	0.05
Santoquin	0.02	0.02	0.02	0.02	0.02
Phytase ^5^	0.01	0.01	0.01	0.01	0.01
Xylanase ^6^	0.03	0.03	0.03	0.03	0.03
Titanium dioxide	0.50	0.50	0.50	-	-
Inert filler ^7^	1.19	0.62	0.05	0.05	0.05
Calculated composition, % unless noted otherwise					
AME_n_, kcal/kg	3008	3058	3108	3108	3175
CP	22.00	22.00	22.00	20.00	19.00
Digestible lysine	1.18	1.18	1.18	1.05	0.95
Digestible TSAA	0.89	0.89	0.89	0.80	0.74
Digestible threonine	0.77	0.77	0.77	0.69	0.65
Calcium	0.90	0.90	0.90	0.84	0.76
Available *p*	0.45	0.45	0.45	0.42	0.38
Analyzed composition, % unless noted otherwise					
Gross energy, kcal/kg	4027	4082	4062	4038	4097
CP	23.15	23.65	22.95	21.55	20.55
Ether extract	5.00	5.41	6.00	4.71	5.35

^1^ DDGS—distiller’s dried grains with solubles; AME_n_—nitrogen-corrected apparent metabolizable energy. ^2^ Supplied the following per kg of diet: manganese, 100 mg; zinc, 100 mg; copper, 10.0 mg; iodine, 1.0 mg; iron, 50 mg; magnesium, 27 mg. ^3^ Supplied the following per kg of diet: vitamin A, 30,863 IU; vitamin D3, 22,045 ICU; vitamin E, 220 IU; vitamin B12, 0.05 mg; menadione, 6.0 mg; riboflavin, 26 mg; d-pantothenic acid, 40 mg; thiamine, 6.2 mg; niacin, 154 mg; pyridoxine, 11 mg; folic acid, 3.5 mg; biotin, 0.33 mg. ^4^ Supplied 0.12 mg of selenium per kg of diet. ^5^ Optiphos^®^ (Huvepharma Inc., Peachtree City, GA, USA) provided 250 FTU/kg of diet. ^6^ Hostazym^®^ X 250 (Huvepharma Inc., Peachtree City, GA, USA) provided 250 FTU/kg of diet. ^7^ Solka-Floc (Cellulose International Fiber Corporation, North Tonawanda, NY, USA) was used as the inert filler to provide space for Clinacox^®^ (Huvepharma Inc., Peachtree City, GA, USA) and the addition of soybean oil.

**Table 2 animals-14-01524-t002:** Effects of coccidiosis vaccination and starter diet energy concentrations on oocyst per gram of excreta sample (OPG) and plasma carotenoid concentration (µg/mL) at 11 d post-hatch ^1,2^.

Item	Standard		Moderate		High		SEM	*p*-Values ^3^		
	ACD	VAC	ACD	VAC	ACD	VAC		Energy	CCP	Interaction
OPG ^4^	0.016 (268)	0.086 (8113)	0.026 (1416)	0.091 (9826)	0.021 (567)	0.090 (14,746)	0.0074	0.543	<0.001	0.908
Plasma carotenoids	1.96	1.17	1.69	1.05	1.86	1.05	0.191	0.557	0.001	0.876

^1^ Values are LSMeans of 18 or 19 replicate pens for OPG and 7 replicate pens for plasma carotenoids. ^2^ Abbreviations: ACD—birds unvaccinated and given an in-feed anticoccidial drug; VAC—vaccinated, birds were given a commercial dose of vaccine on the day of hatch. CCP—coccidiosis control program. Standard = 3008 kcal/kg; moderate = 3058 kcal/kg; high = 3108 kcal/kg of nitrogen-corrected apparent metabolizable energy. ^3^ Overall ANOVA *p*-values for the effects of energy, vaccination, and their interaction. ^4^ Statistical analysis was performed on arcsin square root transformed data for OPG. Raw means are presented in parentheses.

**Table 3 animals-14-01524-t003:** Effects of coccidiosis vaccination and starter diet energy concentrations on broiler growth performance and apparent ileal digestibility of nutrients and energy in broilers at 11 d post-hatch ^1,2^.

Item	Standard		Moderate		High		SEM	*p*-Values ^3^		
	ACD	VAC	ACD	VAC	ACD	VAC		Energy	CCP	Interaction ^4^
Performance from 0 to 11 d post-hatch										
Body weight gain, kg/bird	0.288	0.281	0.295	0.277	0.283	0.276	0.004	0.277	0.002	0.316
Feed intake, kg/bird	0.349	0.340	0.353	0.339	0.334	0.341	0.005	0.127	0.182	0.061
FCR	1.219	1.232	1.200	1.247	1.189	1.244	0.014	0.817	0.001	0.270
Apparent ileal digestibility at 11 d post-hatch										
Dry matter, %	69.1	63.4	70.4	64.7	71.4	65.8	1.10	0.077	0.001	0.999
Nitrogen, %	80.3	76.2	82.0	77.2	83.4	79.1	0.88	0.003	0.001	0.767
Ether extract, %	92.3	89.2	90.6	84.7	90.9	81.3	1.86	0.024	0.001	0.142
Energy, %	72.3	66.6	73.5	67.5	74.8	68.4	1.14	0.116	0.001	0.955
IDE ^5^, kcal/kg	3432	3125	3530	3239	3581	3271	54	0.014	0.001	0.980

^1^ Values are LSMeans of 19 replicate pens for broiler performance and 6 or 7 replicate pens for nutrient digestibility. ^2^ Abbreviations: ACD—birds unvaccinated and given an in-feed anticoccidial drug; VAC—vaccinated, birds were given a commercial dose of vaccine on the day of hatch. CCP—coccidiosis control program. Standard = 3008 kcal/kg; moderate = 3058 kcal/kg; high = 3108 kcal/kg of nitrogen-corrected apparent metabolizable energy. ^3^ Overall ANOVA *p*-values for the effects of energy, vaccination, and their interaction. ^4^ In the case of an interaction, an asterisk (*) denotes statistical significance (*p* < 0.05) due to CCP within a dietary energy level based on a single degree of freedom orthogonal contrasts. ^5^ IDE—ileal digestible energy.

**Table 4 animals-14-01524-t004:** Effects of coccidiosis vaccination and starter (0 to 18 d) diet energy concentrations on broiler growth performance ^1,2^.

Item	Standard		Moderate		High		SEM	*p*-Values ^3^		
	ACD	VAC	ACD	VAC	ACD	VAC		Energy	CCP	Interaction ^4^
0 to 18 d post-hatch										
Body weight gain, kg/bird	0.700	0.681	0.711	0.659	0.703	0.667	0.009	0.740	0.001	0.206
Feed intake, kg/bird	0.899	0.879	0.899	0.859	0.881	0.881	0.012	0.671	0.044	0.233
FCR	1.290	1.316	1.279	1.328 *	1.264	1.340 *	0.008	0.981	0.001	0.006
0 to 31 d post-hatch										
Body weight gain, kg/bird	2.096	2.094	2.115	2.046	2.112	2.049	0.021	0.703	0.007	0.178
Feed intake, kg/bird	2.972	2.956	2.963	2.919	2.975	2.965	0.029	0.579	0.330	0.814
FCR	1.445	1.443	1.415	1.479 *	1.416	1.459 *	0.009	0.507	0.001	0.001
0 to 43 d post-hatch										
Body weight gain, kg/bird	3.419	3.439	3.424	3.384	3.429	3.392	0.028	0.628	0.377	0.447
Feed intake, kg/bird	5.392	5.406	5.363	5.350	5.409	5.395	0.434	0.505	0.892	0.936
FCR	1.603	1.595	1.579	1.627 *	1.584	1.602	0.008	0.381	0.001	0.002

^1^ Values are LSMeans of 12 replicate pens. ^2^ Abbreviations: ACD—birds unvaccinated and given an in-feed anticoccidial drug; VAC—vaccinated, birds were given a commercial dose of vaccine on the day of hatch. CCP—coccidiosis control program. Standard = 3008 kcal/kg; moderate = 3058 kcal/kg; high = 3108 kcal/kg of nitrogen-corrected apparent metabolizable energy. ^3^ Overall ANOVA *p*-values for the effects of energy, vaccination, and their interaction. ^4^ In the case of an interaction, an asterisk (*) denotes statistical significance (*p* < 0.05) due to CCP within a dietary energy level based on single degree of freedom orthogonal contrasts.

**Table 5 animals-14-01524-t005:** Effects of coccidiosis vaccination and starter diet energy concentrations on broilers processed at 44 d post-hatch ^1,2^.

Item	Standard		Moderate		High		SEM	*p*-Values ^3^		
	ACD	VAC	ACD	VAC	ACD	VAC		Energy	CCP	Interaction ^4^
Hot Carcass										
Weight, kg	2.541	2.530	2.533	2.506	2.557	2.520	0.029	0.777	0.301	0.900
Yield, %	75.12	74.73	75.26	75.05	75.40	74.87	0.175	0.351	0.012	0.672
Fat Pad										
Weight, kg	0.034	0.032	0.034	0.035	0.037	0.034	0.002	0.203	0.404	0.360
Yield, %	1.00	0.95	0.99	1.05	1.08	1.02	0.043	0.160	0.533	0.210
Chilled Carcass										
Weight, kg	2.575	2.565	2.569	2.538	2.592	2.555	0.029	0.766	0.284	0.895
Yield, %	76.13	75.75	76.34	76.01	76.43	75.94	0.176	0.312	0.008	0.894
Breast										
Weight, kg	0.683	0.680	0.684	0.673	0.688	0.680	0.010	0.757	0.202	0.965
Yield, %	20.46	20.04	20.31	20.12	20.23	20.15	0.150	0.925	0.063	0.510
Tender										
Weight, kg	0.132	0.134	0.130	0.130	0.133	0.132	0.002	0.196	0.937	0.624
Yield, %	3.91	3.91	3.86	3.89	3.90	3.92	0.030	0.455	0.495	0.848
Leg Quarters										
Weight, kg	0.785	0.788	0.779	0.781	0.789	0.792	0.010	0.517	0.740	0.999
Yield, %	23.14	23.26	23.15	23.41	23.30	23.54	0.133	0.248	0.053	0.829
Wings										
Weight, kg	0.269	0.270	0.273	0.263 *	0.277	0.265 *	0.002	0.489	0.001	0.023
Yield, %	7.96	7.97	8.12	7.89	8.08	7.89	0.617	0.797	0.008	0.094

^1^ Values are LSMeans of eight replicate pens. ^2^ Abbreviations: ACD—birds unvaccinated and given an in-feed anticoccidial drug; VAC—vaccinated, birds were given a commercial dose of vaccine on the day of hatch. CCP—coccidiosis control program. Standard = 3008 kcal/kg; moderate = 3058 kcal/kg; high = 3108 kcal/kg of nitrogen-corrected apparent metabolizable energy. ^3^ Overall ANOVA *p*-values for the effects of energy, vaccination, and their interaction. ^4^ In the case of an interaction, an asterisk (*) denotes statistical significance (*p* < 0.05) due to CCP within a dietary energy level based on single degree of freedom orthogonal contrasts.

## Data Availability

Data are available by contacting the corresponding author.

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
