# Peer review of "Effects of Starter Diet Energy Concentration on Nutrient Digestibility and Subsequent Growth Performance and Meat Yields of Broilers under Two Coccidiosis Control Programs"

_animals, 2024, doi:10.3390/ani14111524_

Round 1
Reviewer 1 Report
Comments and Suggestions for Authors
The manuscript explores the effects of starter diet energy concentration on nutrient digestibility and subsequent growth performance and meat yields of broilers under two coccidiosis control programs. The main problems are as follows:
1. In the experimental design, are coccidia oocysts present in the environment? Why was there no control group other than VAC and ACD, since coccidia oocysts were detected in both the VAC and ACD groups (Table 2).
2. Why were male broiler by-product chicks from a Cobb female line chosen for the experiment rather than the commercial generation males and females?
3. Based on the description of the sentence in Line 115, "18 to 31 d " should be changed to"19 to 31 d ", and"31 to 43 d " should be changed to "32 to 43 d " in Line 100. In the Line 247 and 254, "44 d " should be changed to "43 d " for consistency with the description of the Material and Methods.
4. Line 186-187: Is the description "with no independent or interactive effects" correct? Independent should be changed to dependent?
5. Like "Experimental (0 -18 d)", the " Common Grower " and " Common Finisher " in Table 1 should be labeled with start and finish times.
6. Please explain why did the authors chose to test the indicators at 11d?
Comments on the Quality of English LanguageMinor editing of English language required
Reviewer 2 Report
Comments and Suggestions for Authors
Article
Effects of starter diet energy concentration on nutrient digestibility and subsequent growth performance and meat yields of broilers under two coccidiosis control programs
Well-written manuscript
Please include information on the decision to use that energy levels
Please explain better the animals used to ileal digesta collection. Not easy to understand the MM. Two experiments? You can improve this also in the title of result tables
Table 3. please review the P values for FI
Table 4 and 5. not clear the asterisk usage. You can also open the interaction and present all treatments separated for FCR
L339 - starter period (please include ages)
Reviewer 3 Report
Comments and Suggestions for Authors
In general, the paper is well-written and comprehensive. Consider applying only minor changes, more specifically:
· At the end of Abstract, it is recommended to summarize about the negative effect of increasing dietary lipid content in starter diet of broilers vaccinated for coccidiosis, similarly as presented in L21-23.
· In L74, explain the term “processing characteristics”
· In L127-129 consider adding a brief description of the methods, besides citation.
· L149: Add a right parenthesis [)] and a period (.)
· L196: It is recommended to use the full name instead of the abbreviation (IDE) in the title.
· In Table 4 title, define which is the starter period (day number).
Reviewer 4 Report
Comments and Suggestions for Authors
Congratulations to the authors for the excellent work, with good writing and information relevant to science. When I first read the manuscript to understand the purpose, I saw that it had publication potential; but I had a big problem understanding the different results in the tables. In the second reading, the idea is to clarify certain doubts, add information and improve writing. Necessary adjustments follow:
1) In the abstract section, the last sentence (conclusion) is very general, which doesn't actually say much. Rewrite.
2) The introduction section was very good, but I believe it could be even better with well-defined hypothesis(es). I also missed making it clear, because in the initial phase of creation this care is so important and justifies this research.
3) The M&M section, two things:
a) Is the excreta used to count oocysts a pool from the pen? needs more details on this collection and sampling process.
b) the statistical data analysis section explains everything how it was done; but it does not make clear the previous tests that should have been carried out; such as: normality test. Oocyst counts are unlikely to be normal in my experience, so I ask, were these tests performed? Has the data been transformed? This needs to be very detailed.
4) I was very happy to see that the authors took care to analyze the diets; presenting calculated and analyzed data in the table. It is often used to calculate digestibility coefficients and is not presented.
5) The presentation of results in tables is appropriate; however, the use of an asterisk (*) to show statistical differences does not make it clear who differs from whom. Authors should look for another way to make this data clear, many authors prefer different letters. If the asterisk is maintained, much better observation is needed about what should be seen; remembering that tables and figures need to be self-explanatory.
6) in the discussion section, the writing leads to the understanding that the work was as expected and the results are very clear. But in this type of work, it is difficult in my opinion to have all these guarantees here. In these situations, I think it is important to include sentences or paragraphs with the limitations of the research or results; to signal the beds that they should be careful when interpreting.
6) The conclusion must answer the objective, I noticed that this was partially done here. Review.
Round 2
Reviewer 4 Report
Comments and Suggestions for Authors
The paper is good; with very interesting results. The authors made the requested corrections; clarified the doubts. The footer of the tables was also good, with details and explanations. Statistical writing has been expanded and detailed. The tables need to be better formatted, but I believe this will be done in the article editing process. I recommend the publication.